# Probing the Effect of Acidosis on Tether-Mode Mechanotransduction of Proprioceptors

**DOI:** 10.3390/ijms241612783

**Published:** 2023-08-14

**Authors:** Yuan-Ren Cheng, Chih-Hung Chi, Cheng-Han Lee, Shing-Hong Lin, Ming-Yuan Min, Chih-Cheng Chen

**Affiliations:** 1Department of Life Science, National Taiwan University, Taipei 10090, Taiwan; infinityster@gmail.com; 2Institute of Biomedical Sciences, Academia Sinica, Taipei 11529, Taiwan; 3Neuroscience Program of Academia Sinica, Academia Sinica, Taipei 11529, Taiwan

**Keywords:** acidosis, APETx2, ASIC3, mechanotransduction, proprioceptors, SDNS

## Abstract

Proprioceptors are low-threshold mechanoreceptors involved in perceiving body position and strain bearing. However, the physiological response of proprioceptors to fatigue- and muscle-acidosis-related disturbances remains unknown. Here, we employed whole-cell patch-clamp recordings to probe the effect of mild acidosis on the mechanosensitivity of the proprioceptive neurons of dorsal root ganglia (DRG) in mice. We cultured neurite-bearing parvalbumin-positive (Pv+) DRG neurons on a laminin-coated elastic substrate and examined mechanically activated currents induced through substrate deformation-driven neurite stretch (SDNS). The SDNS-induced inward currents (*I*_SDNS_) were indentation depth-dependent and significantly inhibited by mild acidification (pH 7.2~6.8). The acid-inhibiting effect occurred in neurons with an *I*_SDNS_ sensitive to APETx2 (an ASIC3-selective antagonist) inhibition, but not in those with an *I*_SNDS_ resistant to APETx2. Detailed subgroup analyses revealed *I*_SDNS_ was expressed in 59% (25/42) of Parvalbumin-positive (Pv+) DRG neurons, 90% of which were inhibited by APETx2. In contrast, an acid (pH 6.8)-induced current (*I*_Acid_) was expressed in 76% (32/42) of Pv+ DRG neurons, 59% (21/32) of which were inhibited by APETx2. Together, ASIC3-containing channels are highly heterogenous and differentially contribute to the *I*_SNDS_ and *I*_Acid_ among Pv+ proprioceptors. In conclusion, our findings highlight the importance of ASIC3-containing ion channels in the physiological response of proprioceptors to acidic environments.

## 1. Introduction

Proprioception is an awareness that underlies body–limb coordination and determines both unconscious reflexes and conscious motor tasks [1,2]. Specialized mechanoreceptors innervate muscle spindles (MSs) and Golgi tendon organs (GTOs), which are collectively called “proprioceptors” [3]. Analysis of molecular identity in proprioceptors has revealed that calcium-binding proteins parvalbumin (Pv), Runt-domain transcription factor (RUNX3), and vesicular glutamate transporter (VGULT2) are expressed predominantly in MS afferent proprioceptors [3,4,5,6,7,8]. These MS afferents consist of mechanically activated channels (Macs) that mediate the mechanotransduction of the peripheral nerves into the central nervous system. Based on the channel gating mechanism, Macs can be gated by either membrane tension change (bilayer model) or tethering proteins that link to the extracellular matrix and/or cytoskeletons (tether model) [9]. Although plenty of explanations about how Macs transduce the mechanical strain into electrical signals have been proposed, there are a limited number of methods for describing neurosensory mechanotransduction at the cellular level [10]. A common approach to assessing Mac activity is to directly apply force on the cell soma via glass pipette indentation, through which Piezo1 and Piezo2 have been identified as non-selective cation channels in whole-cell patch-clamp recordings [11]. Piezo2 is expressed in parvalbumin-positive (Pv+) dorsal root ganglion (DRG) proprioceptive neurons, and the conditional knockout of Piezo2 in Pv+ DRG neurons shows that Piezo2 is the major contributor to rapid adaptive mechanically activated responses to the direct soma indentation in proprioceptors [12]. Meanwhile, the intermediary adaptive mechanically activated responses are slightly upregulated by Piezo2 ablation, which suggests that Piezo2-independent Macs may also be involved in the mechanotransduction of proprioceptors [13].

Contrarily, to identify the tether-mode Macs of proprioceptors, we developed a method in which neurites on an extracellular matrix (ECM)-coated elastomeric substrate are stretched by imposing localized substrate deformation, so that the electrical responses of tether-mode Macs can be recorded via whole-cell patch clamping [14]. This method can provide a deeper understanding of mechanotransduction at the subcellular level of proprioceptors [14]. The targeted knockout of acid-sensing ion channel 3 (Asic3) on Pv+ DRG neurons disrupts MS afferent sensitivity and impairs substrate deformation-driven neurite stretch (SDNS) mechanotransduction, but not the soma indentation-induced mechanotransduction [15]. Behaviorally, Asic3 knockout mice maintain normal locomotion function as shown in open-field activity and rotarod performance, but the animals exhibit a proprioceptive deficit in the grid walking test in the dark and a balance task using a beam with a size of less than 6 fields [15]. Together, these findings suggest that ASIC3 is involved in mechanosensing associated with the tether-mode mechanotransduction of proprioceptors [15,16,17]. Since ASIC3 has a dual function of acid sensing and mechanosensing, how ASIC3-dependent proprioception is influenced by tissue acidosis should be a topic of scrutiny.

The fine adjustment of body coordination relies on precise proprioceptive input, which is crucial to achieving balance. Balance can be impaired by fatigue, intermittent training, and intensive exercise [1,18,19,20,21,22]; however, the molecular and cellular mechanisms underlying fatigue-associated balance disturbances remain under-explored. Previous microdialysis studies on humans and rodents have demonstrated that the levels of muscle interstitial lactate and protons substantially increase during exercise as compared with rest, and that they tend to increase further with exercise intensity [23]. Moreover, when the rat hindlimb is subjected to a supramaximal stimulation, H^+^ increases earlier than lactate, which leads a minor acidic level (pH = 7.2) in the muscle [24]. When human subjects undergo intermittent training, the H^+^ released at exhaustion is significantly higher in the legs of the trained subjects than in those of the untrained subjects at the 8th week of training, and at the 30th week, the blood pH and muscle pH decrease to 7.13 and 6.82, respectively, in the trained legs [25]. It can be concluded from these results that exercise and fatigue affect the homogenous blood flow and muscle metabolism. However, it is still unknown how this micro-environment affects proprioception and leads to balance impairment.

This study sought to explore how the micro-environment of fatigued muscles influences proprioception. Specifically, we used whole-cell patch-clamp recordings of Pv+ proprioceptors to probe how mild acidosis modulates the tether-mode mechanotransduction of proprioceptors.

## 2. Results

### 2.1. Pv-Expressing DRG Nerve Terminals Are Proprioceptive Fibers Innervating MSs and GTOs

To investigate the effect of acidosis on the neurosensory mechanotransduction of proprioceptors, we used a genetic model to label the proprioceptors of DRG. We also used the Cre::LoxP reporting system to characterize the muscle afferents in the soleus muscle of Pv-Cre::CAG-cat-EGFP mice. In the immunofluorescence study, we examined six Pv-expressing muscle afferents and found that they can be identified as one of three types of proprioceptive nerve terminals, namely groups Ia and II muscle spindle afferents and group Ib Golgi tendon organ afferents in the hind limb soleus muscle (Figure 1). On the muscle edge, Pv+ eGFP immunoreactivity was observed in free nerve terminals in the soleus muscle, which are the GTO afferents or group Ib proprioceptors (Figure 1a), and in the annunospiral fibers and the free nerve endings in the intrafusal bag, which are group Ia and group II proprioceptive nerve terminals, respectively (Figure 1b,c). All three of these types of DRG nerve terminals were neurofilament heavy chain-positive, which indicates myelinated nerve fibers.

### 2.2. Acidic Environment Attenuates the Mechanically Activated Currents Induced by SDNS

We used whole-cell patch clamping to record the mechanically activated currents of SDNS in tdTomato+ neurons isolated from Pv-Cre::tdTomato mice (Figure 2a). Laminin-coated polydimethylsiloxane (PDMS) served as an elastic basement for Pv+ neurite growth. For each recording, action potentials (APs) and rheobase were first analyzed in the normal artificial cerebral spinal fluid (ACSF). Through this procedure, we identified at least three proprioceptor subtypes, based on their firing patterns, as having single, burst, and tonic APs (Table 1). Then, the normal ACSF was replenished with tetrodotoxin (TTX, 300 nM)-containing ACSF for mechano-clamping. To deliver a series of mechanical stimuli to the PDMS surface, a 40 µm-wide glass electrode with an interior opening of less than 1 MOhm was used as an indentation pipette and connected to a 3-axis mechanical actuator (Figure 2b). After mechano-clamping, we used a six-channel valve controller and three-barrel pipette to apply acidic perfusion (Figure 2c). The perfusion area covered the soma and neurites of interest. The tip of the perfusion pipette was 1 cm away from the recording soma (Figure 2d). One of the three-barrel pipettes constantly delivered the exact pH of ACSF as a bathing solution. These procedures were meant to reduce the effect of mechanical stress from perfusion. The six-channel valve controller regulated the inlet of acidic ACSF (Figure 2e).

The Pv+ neurites, which were tightly bound to the PDMS surface, were stretched by substrate deformation via a vertical pipette indentation (Figure 2f,g). We performed whole-cell patch clamping to measure the substrate deformation-driven neurite stretch-induced currents (*I*_SDNS_) of neurite-bearing Pv+ DRG neurons and examined the effect of acidosis (pH 6.8) on the *I*_SDNS_ (Figure 3a). We first identified seven Pv+ neurons that expressed *I*_SDNS_ in response to mechanical stimuli with an indentation greater than 100 µm in the neutral pH 7.4 ACSF, and the *I*_SDNS_ was significantly attenuated when the local pH was dropped to 6.8 (Figure 3b,c). The *I*_SDNS_ was force-dependently increased as the indentation depth increased in both neutral and acidic pH (Figure 4a–c). In the pH 7.4 ACSF, the *I*_SDNS_ peak amplitudes were (expressed in pA) 0.52 ± 1.63, −2.02 ± 1.80, −6.01 ± 2.31, −13.98 ± 2.27, and −18.96 ± 4.67 in response to indentation depths of 25, 50, 75, 100, and 125 µm, respectively (Figure 4d). In the pH6.8 ACSF, the SDNS currents were only detectable with amplitudes of −3.42 ± 1.58 pA and −7.08 ± 2.66 pA with indentation depths of 100 and 125 µm, respectively. 

Although there were notable differences among the AP firing patterns, an analysis of the *I*_SDNS_ current amplitudes at pH 7.4 in relation to cell size, rheobase, and AP firing patterns revealed no correlation between these factors (Appendix A). This finding indicates that the current amplitude is not associated with the properties of the AP.

To investigate the effect of acidosis on Pv+ proprioceptors in depth, we examined the *I*_SDNS_ in different pH environments ranging from 6.8 to 7.4 (Figure 5a) using 100 µm indentations. The bath solution was pre-conditioned to pH 8.0 ACSF with TTX, and the perfusion solution was applied to the recording cell two minutes before indentation. The *I*_SDNS_ current amplitudes were pH-dependently attenuated as the pH dropped from 7.2 to 6.8 (Figure 5b). Quantitative analyses showed that the average current amplitude of the *I*_SDNS_ was −13.79 ± 1.85 pA at pH 7.4 (*n* = 32), which was significantly reduced to −9.32 ± 1.24 pA at pH 7.2, −7.16 ± 1.36 pA at pH 7.0, and −4.97 ± 1.07 pA at pH 6.8 (Figure 5c). Thus, the inhibitory effects of acidosis on the *I*_SDNS_ were more prominent in the Pv+ proprioceptors with single APs or tonic APs than in the proprioceptor subtype with burst APs (Figure 5d–f). 

### 2.3. Role of ASIC3 in Tether-Mode Mechanotransduction during Acidosis

Previous studies have shown that ASIC3 is not only a sensitive acid sensor for mild acidosis but also a mechanic sensor involved in the tether-mode mechanotransduction of Pv+ DRG proprioceptors [15,26]. To explore how acidosis modulates the ASIC3-dependent *I*_SDNS_, we designed a protocol to determine the relationship of ASIC3-dependent currents (*I*_ASIC3_) with the *I*_SDNS_ and acid-induced current (*I*_Acid_) in different Pv+ DRG proprioceptors (Figure 6a). We tested the effects of APETx2 (2 µM), a selective antagonist for ASIC3 [27], on the *I*_SDNS_ in both pH 7.4 and pH 6.8 conditions, and on the *I*_Acid_ in pH 6.8 conditions. In pH 7.4 conditions, the *I*_SDNS_ was expressed in 60% (25/42) of Pv+ DRG neurons, and the current was inhibited by APETx2 in 92% (23/25) of *I*_SDNS_-expressing neurons (Figure 6b, Appendix A). Quantitative analyses revealed that APETx2 significantly reduced the *I*_SDNS_ in Pv+ DRG neurons with single or burst APs, but not in those with tonic APs (Figure 7a–d). In addition, among 25 *I*_SDNS_-expressing Pv+ DRG neurons, pH 6.8 acidosis attenuated the APETx2-sensitive *I*_SDNS_ (*n* = 23) but had no effect on the APETx2-resistant *I*_SDNS_ (*n* = 2) (Figure 7e). Intriguingly, among these 23 APETx2-sensitive *I*_SDNS_-expressing Pv+ neurons, acid perfusion of pH 6.8 only induced *I*_Acid_ in 19 neurons, where the *I*_Acid_ was inhibited by APETx2 in 11 of these neurons and defined as *I*_ASIC3_ (Figure 7f,g). Mild acidosis of pH 6.8 significantly attenuated the *I*_SDNS_ in both groups of neurons, respectively expressing *I*_SDNS_/*I*_Acid_ and *I*_SDNS_ only (Figure 7f). For those 19 neurons with *I*_SDNS_/*I*_Acid_, pH 6.8 acidosis significantly attenuated the *I*_SDNS_ in neurons expressing *I*_ASIC3_ (*n* = 11), but not in neurons without *I*_ASIC3_ (*n* = 9) (Figure 7g). All these results indicate a strong association between the inhibitory effect of acidosis on the *I*_SDNS_ and on whether the *I*_SDNS_ or *I*_Acid_ can be inhibited by APETx2. 

Regarding the acid sensitivity, 76% (32/42) of Pv+ DRG neurons expressed the *I*_Acid_ in response to the pH 6.8 acid stimulation (Figure 6b). Interestingly, the *I*_Acid_ was expressed in 84% (21/25) of *I*_SDNS_-expressing neurons and 65% (11/17) of *I*_SDNS_-negative neurons. The average peak amplitudes of the *I*_Acid_ showed no difference between Pv+ DRG neurons with and without the *I*_SDNS_ (Figure 7h) or between *I*_ASIC3_-positive and *I*_ASIC3_-negative groups (Figure 7i). Based on the expression of the *I*_SDNS_, *I*_Acid_, and *I*_ASIC3_, we organized these 42 Pv+ DRG neurons into 7 subgroups to highlight the heterogeneous functionality among Pv+ DRG proprioceptors (Figure 8). 

## 3. Discussion

In this study, the SDNS approach was employed to probe the effect of mild acidosis on the tether-mode mechanotransduction of proprioceptive neurites. One advantage of this approach is that it delivers a specific mechanical mode emulating physiologically relevant conditions, in which the proprioceptive terminals of MSs or GTOs are stretched during muscle contraction [10]. It was found that 59% (25/42) of Pv+ DRG proprioceptors expressed the *I*_SDNS_, and that ASIC3 was the major molecular determinant contributing to the *I*_SDNS_. This finding is consistent with previous studies that showed amiloride-sensitive sodium channels were the main mechanically activated ion channels in response to MS stretching in ex vivo models [28]. Moreover, using a series of experimental designs, this study showed that mild acidosis significantly attenuated the *I*_SDNS_ in most ASIC3-positive proprioceptors. The *I*_SDNS_-expressing Pv+ DRG proprioceptors were functionally heterogeneous in terms of their AP profiles, sensitivity to neurite stretching, and acidosis. In particular, not all *I*_SDNS_-expressing Pv+ DRG proprioceptors expressed the *I*_Acid_ in response to the pH 6.8 ACSF. Of the 23 *I*_SDNS_-expressing Pv+ proprioceptors, 19 expressed the *I*_Acid_, but the *I*_Acid_ was only inhibited by the ASIC3-selective antagonist APETx2 in 61% (11/18) of Pv+ proprioceptors, suggesting that the ASIC3-containing ion channels are highly heterogenous among *I*_SDNS_-expressing proprioceptors.

The fact that mild acidosis attenuated the *I*_SDNS_ in proprioceptive neurons provides valuable insights into the impairment of balance caused by fatigue or intermittent exercise [18]. Proprioceptive neurons play a crucial role in providing sensory information about body position and movement for the central nervous system. The attenuation of the *I*_SDNS_ due to mild acidosis suggests that acidosis could disrupt the mechanotransduction of proprioceptors, potentially leading to a diminished ability to accurately sense and respond to mechanical stimuli. This impairment in proprioceptive function could have significant consequences for maintaining balance and coordinating movements during exercise or in situations involving metabolic acidosis.

The inhibitory effect of mild acidosis on the *I*_SDNS_ appears to have been highly associated with the expression of ASIC3, as the acid-inhibiting effect occurred in the *I*_SDNS_ sensitive to APETx2 inhibition but not in that resistant to APETx2 (Figure 7e). APETx2 is a 42-amino-acid peptide isolated from the sea anemone *Anthopleura elegantissima*, known to inhibit ASIC3 homomeric channels and ASIC3-containing heteromeric channels [27]. Although APETx2 reversibly inhibits rat ASIC3 without affecting ASIC1a, ASIC1b, or ASIC2a, it exerts stronger inhibitory effects on the *I*_Acid_ of homomeric ASIC3 or heteromeric ASIC2b+3 than on that of ASIC1a+3 and ASIC1b+3 channels [27]. In addition, APETx2 was more effective in inhibiting the *I*_SDNS_ than inhibiting the *I*_Acid_, as APETx2 failed to inhibit the *I*_Acid_ in 35% (8/23) of *I*_SDNS_-expressing Pv+ proprioceptors (Figure 6b). Interestingly, mild acidosis did not significantly attenuate the *I*_SDNS_ in neurons expressing an APETx2-resistant *I*_Acid_ (Figure 7g). This suggests that mild acidosis might differentially influence the *I*_SDNS_ between Pv+ proprioceptors expressing an APETx2-sensitive *I*_Acid_ (possibly mediated by ASIC3 or ASIC2b+3) and those expressing an APETx2-resistant *I*_Acid_ (possibly mediated by ASIC1a+3 or ASIC1b+3). In brief, the composition of heteromeric ASIC3-containing channels may determine the effects of mild acidosis on proprioceptive nerve activity during muscle contraction. Further studies should be conducted to understand the functional properties and specific roles of different heteromeric ASIC3 channels in proprioceptive neurons, as well as their modulation of proprioception through acidosis.

Taking into account the pH dependency of ASIC subtypes and heteromeric compositions in heterologous expression systems, the presence of ASIC2a in the channel composition consistently leads to a half-maximum pH activation that is lower than 6.0 [29]. This indicates that ASIC2a-containing channels require a more concentrated acidosis to activate the proton-gated current compared with other subtype combinations. Accordingly, the ASIC2a-containing channels (ASIC2a+3) may account for the Pv+ proprioceptor subtype that expresses an *I*_SDNS_ but not an *I*_Acid_ at pH 6.8 (Figure 6b). Still, ASIC2a-containing channels might also be expressed in some Pv+ proprioceptors without an *I*_SDNS_.

Mild-acidosis-induced structural changes in ASIC channels may underlie the observed attenuation of SDNS currents. Acidosis alters the pH environment surrounding the ASIC channels, potentially impacting their conformation and functional properties. ASIC3, as a homologue to chicken ASIC1a, shares similarities in its activation and desensitization mechanisms in response to low-pH environments [29,30]. Studies on chicken ASIC1a crystal structures have elucidated the structural changes associated with channel activation and desensitization [31]. In the resting state, the thumb domain of ASIC1a moves outward relative to its position in the open and desensitized state, resulting in the expansion of the acidic pocket. Activation involves the closure of the thumb domain into the acidic pocket, the expansion of the lower palm domain, and an iris-like opening of the channel gate. The beta11-12 linker, which separates the upper and lower palm domains, acts as a molecular clutch and undergoes rearrangement to facilitate rapid desensitization, effectively stopping the proton sensing current [32]. Although ASIC3 and ASIC1a may have divergent characteristics, it is reasonable to assume that ASIC3 might exhibit similar structural rearrangements and conformational changes in response to low pH and acidosis. These alterations in the thumb and palm domains, along with the rearrangement of the beta11-12 linker, could potentially influence the mechanical gating properties of ASIC3 channels [31]. As a result, the attenuation of the *I*_SDNS_ observed in mild acidosis may be attributed, at least in part, to the effect of acidosis on the mechanical gating of ASIC3 channels.

We proposed three possible mechanisms to determine how the SDNS response of ASIC3-containing channels in proprioceptive neurons is modulated by acidosis through different mechanisms, depending on the channel composition. In the case of ASIC3 homomeric channels, mild acidosis promoted channel opening, allowing the influx of ions and the initiation of a proton-gated response. However, due to the inherent desensitization properties of ASIC3, the channels were transitioned into a rapid desensitized state, resulting in the attenuation of the SDNS current (Figure 9a). In the case of ASIC3+1a and ASIC3+1b heteromeric channels, mild acidosis also facilitated channel opening, leading to the initiation of a proton-gated response and thus attenuating the SDNS current. However, the presence of ASIC1a or ASIC1b altered the kinetic and pharmacological responses, potentially influencing the magnitude of the SDNS current (Figure 9b). The combination of ASIC3 and ASIC2a in a heteromeric channel resulted in a direct transition into the desensitized state upon exposure to mild acidosis, bypassing the open state of the channel. This indicates that the SDNS response in ASIC3+2a heteromeric channels is primarily mediated by the channels being in a desensitized state without being transitioned into the open state (Figure 9c).

In summary, this study explored the complex interplay between acidosis, channel composition, and the mechanosensitivity of ASIC3-containing channels in proprioceptive neurons. Understanding the specific responses of ASIC3-containing channels to acidosis and their effect on the mechanosensitivity of proprioceptive neurons may provide insight into the regulation of sensory processing and motor control during exercise. Still, this study had some limitations: (1) the SDNS approach only induced an *I*_SDNS_ in 59% of Pv+ proprioceptors, which may have only represented a subgroup of proprioceptors sensitive to the SDNS stimuli; (2) some SDNS-insensitive neurons expressed an *I*_ASIC3_, suggesting they are involved in tether-mode mechanotransduction but require either stronger SDNS stimuli or a different mechanical modality; and (3) we did not know the exact compositions of the ASIC3-containing channels among the *I*_SDNS_-expressing and *I*_SDNS_-insensitive Pv+ proprioceptors. Further research should be undertaken to unravel the precise molecular and biophysical mechanisms underlying the acidosis-induced modulation of ASIC3 channel gating, as well as its implications for proprioceptive function and balance control. Such research may pave the way for the development of targeted therapeutic interventions aimed at improving proprioception in individuals with acidosis-induced balance impairments.

## 4. Materials and Methods

### 4.1. Animals

All the animal protocols followed in this study complied with the Guide for the Use of Laboratory Animals (National Academy of Sciences Press) and were approved by the Institutional Animal Care and Use Committee of the Academia Sinica, Taiwan. Both male and female mice at ages of 12–16 weeks were used. The Pv-Cre::Td (parvalbumin-Cre and tdTomato reporter) and Pv-Cre::CAG-cat-EGFP (parvalbumin-Cre and EGFP reporter) transgenic lines were backcrossed to the C57BL6 background and kept as heterozygotes for all experiments.

### 4.2. Immunostaining of MSs

Procedures for immunostaining and muscle tissue preparation were modified from a picric acid fixative method. The mice were anesthetized with a combination of Zoletil and dexmedetomidine hydrochloride; they were then perfused transcardially with 25 mL of 0.02 M phosphate-buffered saline (PBS, Omics Bio, Taipei, Taiwan) (pH = 7.4, at 4 °C) and cold fixative (4% (*w*/*v*) formaldehyde, 14% (*v*/*v*) saturated picric acid, and 0.1 M PBS (pH = 7.4, at 4 °C)). The animals’ soleus muscles were dissected and post-fixed in the cold fixative solution (4 °C) for 1 h. The fixative was washed out with PBS for a while and cryoprotected in 20% sucrose PBS (pH = 7.3, at 4 °C) for 24 h. After being embedded in the optimal cutting temperature (OCT) compound (Leica Biosystem, Deer Park, IL, USA), the muscle tissues were frozen and sectioned into 12–16 µm thick sections, placed on a gelatin-coated slide glass, and stored at −80 °C before usage. Before immunostaining, the muscle sections were washed three times with PBST (PBS + 0.1% Triton X-100), blocked with PBST containing 3% bovine serum albumin (Sigma-Aldrich, St. Louis, MO, USA) and 5% goat serum (Sigma-Aldrich, St. Louis, MO, USA), and incubated with primary antibodies (rabbit anti-GFP 1:5000 in blocking solution, Abcam, Cambridge, UK; mouse anti-NF-H, 1:2000 in blocking solution, Chemicon, St. Louis, MO, USA) overnight at 4 °C. After PBST washing was performed 3 times, the muscle sections were incubated in the secondary antibodies (Goat anti-rabbit 1:500, Invitrogen, Waltham, MA, USA) for 1 h (at room temperature).

### 4.3. Primary Culture of DRG Neurons 

Mice at the age of 12–16 weeks were sacrificed with CO_2_, and their DRG neurons were collected and cultured in accordance with the procedures described in Cheng et al., (2010) [14] and Lin et al., (2016) [15]. The dissected DRGs were digested with 0.125% collagenase (type I, Sigma-Aldrich, St. Louis, MO, USA) and 2 units/mL dispase II (Sigma-Aldrich, St. Louis, MO, USA) for 30 min at 37 °C. The segregated neurons were triturated using a flame-polished Pasteur pipette and seeded on laminin (Sigma-Aldrich, St. Louis, MO, USA)-coated polydimethylsiloxane (PDMS, UNI WARD, New Taipei City, Taiwan) substrate, which was prepared on a 12 mm coverslip with a base-to-curing-agent ratio of 35:1. Before the DRG neurons were seeded, the PDMS-covered coverslips were exposed to ultraviolet light for 15 min and coated with poly-L-lysine (0.01%, Sigma-Aldrich, St. Louis, MO, USA) for 10 min; then, they were coated with laminin (10 μg/mL, Sigma-Aldrich, St. Louis, MO, USA) for 2 h. The neurons were cultured in a 3.5 cm Petri dish with DMEM plus 10% fetal bovine serum and maintained in an incubator with 5% CO_2_ at 37 °C for 2–3 days. The cultured DRG neurons were then subjected to electrophysiological recordings of acid-induced currents or mechanically activated currents.

### 4.4. Electrophysiology

#### 4.4.1. Whole-Cell Patch-Clamp Recordings

The acid-induced currents (*I*_Acid_) in the Pv-Cre::Td-positive DRG neurons were measured. Six-channel valve gates (white arrow) were attached to the perfusion columns, the other side was conducted to a three-barrel pipette using a six-in one-out device (black arrow), and the acidic artificial cerebral spinal fluid (ACSF) controlled by the six-channel valve controller (Warner, VC-6, Science Products, Hesse, Germany) (red arrow) was applied (Figure 2). Whole-cell patch clamping was conducted on neurite-bearing DRG neurons cultured for 48–72 h. The recording pipette (with a resistance of 6–8 MOhm) was filled with internal solution (in mM, 100 KCl, 2 Na-ATP, 3 Na-GTP, 10 EGTA, 5 MgCl_2_, and 40 HEPES, with the pH adjusted to 7.4 using KOH; osmolality 290–310 mOsm) (Sigma-Aldrich, St. Louis, MO, USA), and the neurons were externally bathed in the ACSF solution (in mM, 130 NaCl, 5 KCl, 1 MgCl_2_, 2 CaCl_2_, 10 glucose, and 20 HEPES, with the pH adjusted to 7.4 or 8.0) (Sigma-Aldrich, St. Louis, MO, USA). The puff pipette was filled with acidic ACSF, whose pH was adjusted to 6.8, 7.0, 7.2, and 7.4. Mechanical stimuli were delivered using a mechanical pipette on an electronically controlled micromanipulator (Scientifica PatchStar Micromanipulator pc8200c, Scientifica, East Sussex, UK). The pipette was microforge-polished (MF-900, Narishige, Tokyo, Japan) to 4–5 MOhm resistance. Loaded with ACSF, the pipette was held at −5 to −10 mV (the leak current slowed to less than −1500 pA) using a Multiclamp 700B (Axon instrument, San Jose, CA, USA) so that we could monitor the contact with the PDMS surface. Once the mechanical pipette was lowered down and touched the PDMS surface, a leak current drop was observed, indicating increased resistance. Thus, we synchronized the timing of the whole-cell recording with the starting point of the PDMS indentation. 

#### 4.4.2. Rheobase Analysis of AP Threshold

The neurons were bathed in an ACSF solution (pH 7.4) and formed by the whole-cell patch. The membrane potential was then held at −70 mV by injecting current in the current clamp mode, and a series of stimuli (from 100 to 2000 pA for 20 sweeps, with a 100 pA increase per step, over a 500 ms duration) was provided to trigger an AP. The rheobase current was determined using the minimal injected current to trigger an AP (Appendix A).

#### 4.4.3. Mechanically Activated Current Recording

After the rheobase current was recorded, the neurons were transferred to be externally bathed in an ACSF solution (pH 8.0) containing TTX (300 nM, Tocris Bioscience, Bristol, UK) to inhibit the voltage-gated sodium channel activity, and the mechanical pipette was placed around the distal neurite terminal in a region on the PDMS surface that satisfied the following criteria: (i) the region did not contain any other cells such as glia or fibroblasts; (ii) the region was at least 100 µm away from the Pv-Cre::Td-selected neuron cell soma; and (iii) the region was 15–25 μm away from the distal terminal (Figure 2b,c). To indent the PDMS substrate, the mechanical pipette was first anchored on the surface as a starting position (p0) and then hoisted on the z-axis for 100 μm (p+100). The indentation of the mechanical pipette was programmed so that the pipette was (i) dropped down on the z-axis for 200 μm (that is, 100 μm below the initial surface plane (p-100)) at a velocity of 1.6 μm ms^−1^; (ii) kept at (p-100) for 1000 ms; and (iii) elevated at the same velocity to (p+100). After pipette indentation was performed on PDMS, the deformation of the substrate stretched local neurites. A series of indentation depths (25, 50, 75, 100, and 125 μm) were tested within the same Pv+ DRG neuron to analyze the force dependency of the SDNS-induced currents (*I*_SDNS_) (Figure 2d).

#### 4.4.4. Mechanically Activated Currents during Acidosis

To study the effect of acidosis on mechanically activated currents, we first recorded the *I*_SDNS_ with indentation depths of 25, 50, 75, 100, and 125 μm in a pH 7.4 bath. A pH 6.8 acidic bath was then applied to the same neurons, and we waited at least 5 min for the acidic ACSF to fully immerse the neurons before measuring the *I*_SDNS_ under the same indentation protocol. This allowed us to evaluate the effect of acidosis on the *I*_SDNS_ (Figure 3).

To further investigate the effect of proton concentrations on different proprioceptor groups, we triggered APs and tested the rheobase of Pv+ neurons in neutral ACSF; then, we used a pH 8.0 TTX-containing ACSF solution for the *I*_SDNS_ recordings. The neurons were perfused with a puff pipette preloaded with TTX-containing ACSF at the pH levels of 6.8, 7.0, 7.2, and 7.4. They were sequentially exposed to acidic challenges from the ACSF with pH levels of 7.2, 7.0, and 6.8 to assess the effect of proton concentrations on the *I*_SDNS_ (with a 100 μm indentation) (Figure 4).

#### 4.4.5. ASIC3 Dependency

To determine whether ASIC3 contributed to the expression of the *I*_Acid_ and *I*_SDNS_ in Pv+ DRG neurons, the ASIC3-selective antagonist APETx2 (2 μM, Alomone Labs, Jerusalem, Israel) was included in the puff pipette containing ACSF with a pH of either 7.4 or 6.8 [15]. After the AP profile measurement, the Pv+ DRG neurons were subjected to a series of recordings of the *I*_Acid_ and *I*_SDNS_ and tested for APETx2 inhibition (Figure 6).

### 4.5. Statistical Analysis 

Data in all figures are presented in the form of mean ± sem. Statistical comparison was performed using Prism (ver. 10.0.2) (ANOVA and *t*-test). A non-parametric two-tailed Mann–Whitney test was used in some electrophysiology experiments where the raw data did not pass the normal distribution or equal variance test. The Holm–Sidak test was employed for post hoc tests with two-way ANOVA and paired one-way ANOVA, because the Holm–Sidak test is more powerful than the Tukey or Bonferroni test. The criterion for a significant difference was set to *p* < 0.05.

## 5. Conclusions

This study focused on the significant role of ASIC3, a proton-gated ion channel with the dual gating properties of acid sensing and mechanosensing, in modulating proprioception under acidic physiological conditions. Understanding the role of ASIC3 in these processes may help to clarify the underlying mechanisms of exercise-induced balance impairments, to develop strategies for mitigating these effects, and to optimize athletic performance. Furthermore, investigating the heterogeneity of Pv+ DRG neurons in the context of ASIC3 and its potential interactions with other ASIC subtypes may improve the understanding of the mechanosensing capabilities of proprioceptive neurons and inform the development of targeted interventions for conditions involving impaired proprioception.

## Figures and Tables

**Figure 1 ijms-24-12783-f001:**
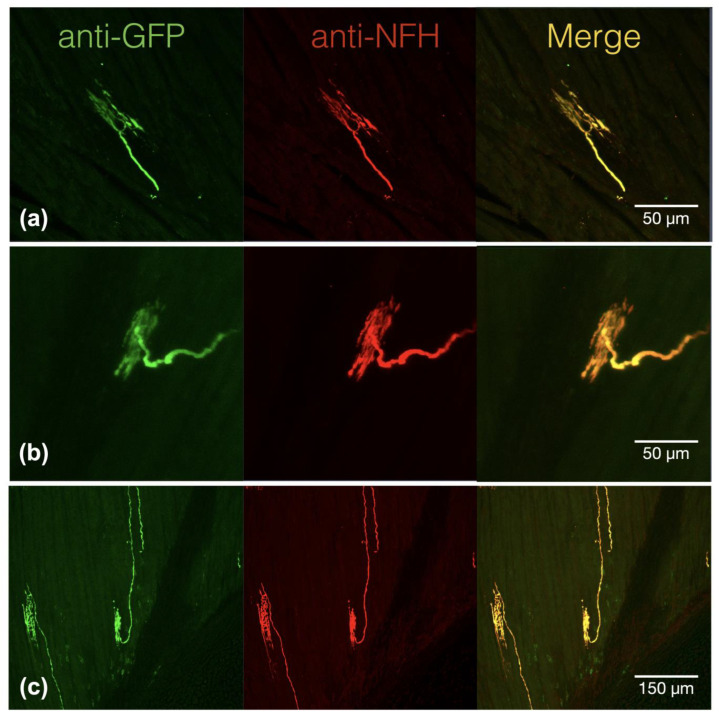
Parvalbumin+ proprioceptive muscle afferents are neurofilament heavy chain immunoreactive muscle afferents with specialized mechanosensing structures. (**a**) Extrafusal muscle afferent showed the co-localized expression of green fluorescence protein (GFP) and neurofilament heavy chain (NFH). (**b**) Intrafusal muscle spindle afferents, which can be classified as type II muscle spindle afferents, displayed co-expression of parvalbumin-triggered GFP and NFH. (**c**) The annunospiral type I muscle spindle afferents in the intrafusal bag also presented strong parvalbumin-triggered GFP and NFH.

**Figure 2 ijms-24-12783-f002:**
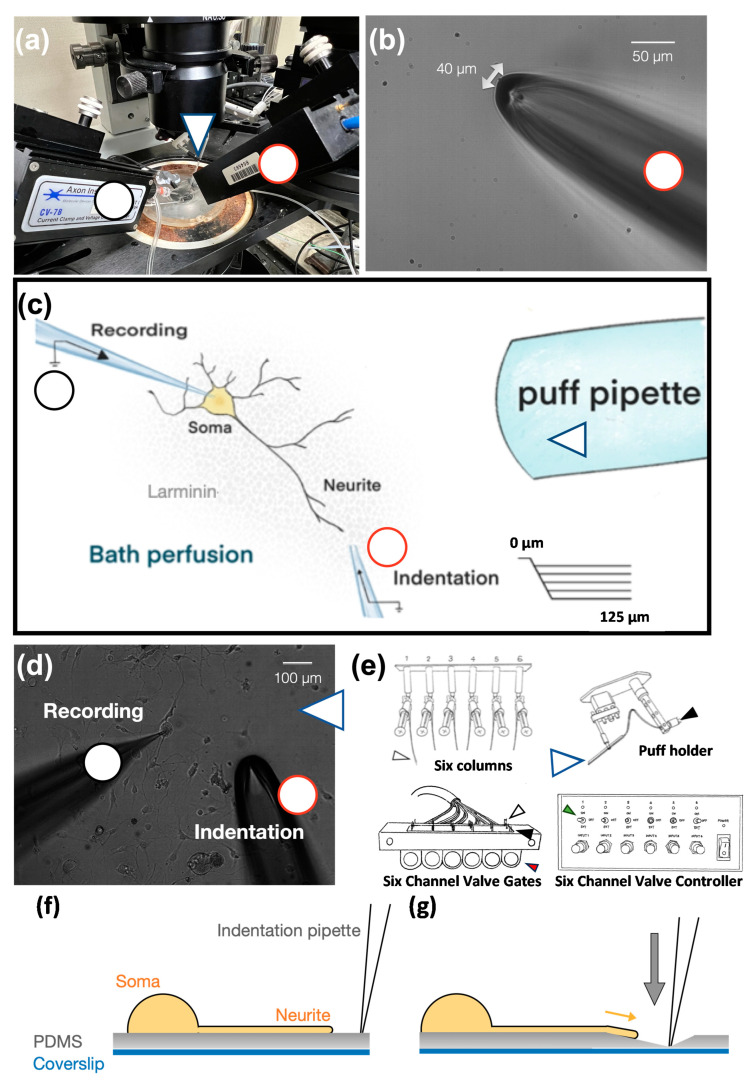
The assembly of the substrate deformation-induced neurite stretch (SDNS) approach. (**a**) Top view of the SDNS mechano-clamp system. The black circle points to the whole-cell patch-clamp recording pipette. The red circle denotes the mechanical actuator-driven indentation pipette. The blue arrow indicates the acidic solution perfusion pipette. (**b**) The mechanical actuator-driven indentation pipette was a hollow borosilicate glass pipette whose tip was forged into a 40 µm blunt opening ending. (**c**) Schematic view of the whole SDNS setup. The neurite-bearing DRG neuron was cultivated on laminin-coated polydimethylsiloxane (PDMS). The neurite process extended to a distal region, and the indentation pipette was positioned 10 µm away to avoid direct contact with the neurite. The recording pipette was sealed with the soma. The perfusion (puff) pipette was far away from the soma to prevent the sheer force against the cell. (**d**) An actual recording and the puff pipette are shown on the right side of the image. (**e**) The six columns contained artificial cerebral spinal fluid (ACSF) with different pH levels to provide an acidic challenge to the cell, controlled by six-channel valve gates. The controller received an electrical signal, opened the gate (green arrow, switch; red arrow, gates), and then moved the puff holder (blue arrows, puff pipette) to deliver acidic challenges (the plastic pipes were linked between the black arrow and white arrow respectively to conduct ACSF). (**f**) The schematic diagram describes the SDNS recording setup. At the beginning, an indentation pipette was close to a target neurite and the surface of the PDMS substrate. (**g**) The schematic diagram depicts the SDNS procedure. After the indentation began, the indentation pipette was inserted into the PDMS substrate (gray arrow indicates the direction of the force), which caused the deformation of the substrate and thus stretched the neurite (yellow arrow indicates the direction of the stretching).

**Figure 3 ijms-24-12783-f003:**
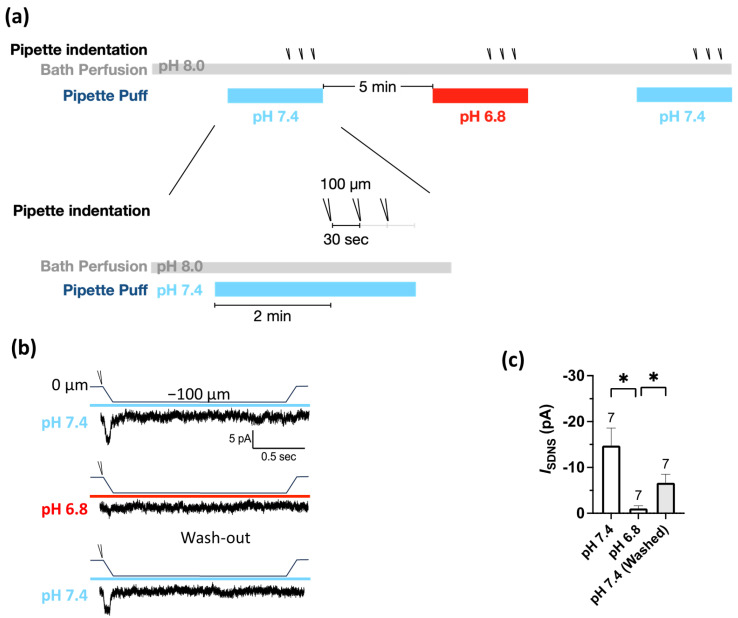
Mild acidosis at pH 6.8 attenuated the substrate deformation-driven neurite stretch (SDNS)-induced currents with a partial reversibility. (**a**) Experimental protocol for testing the effect of acidosis (pH 6.8) on SDNS-induced currents in Pv+ DRG proprioceptors. The bath was a pH 8.0 ACSF, and the perfusion pipette contained either a pH 7.4 or pH 6.8 ACSF during the recordings of SNDS-induced currents. Three repetitive indentations were applied to stretch the neurite to generate SDNS currents; pH 7.4 (colored in light blue) and pH 6.8 (colored in red). (**b**) Three normalized representative SDNS current traces with a 100 µm depth at pH 7.4 or pH 6.8 (with three repetitive indentation traces merged for normalization). (The scale bar indicates 5 pA and 500 ms.) (**c**) The amplitude of *I*_SDNS_ (expressed in pA) during pH 7.4, pH 6.8, and pH 7.4 washings is shown. One-way ANOVA revealed statistically significant differences in the mean test score between the above three groups (*F* (1.245, 7.470) = 9.044, *p* = 0.0153). Multiple comparisons showed that the mean value of *I*_SDNS_ was significantly different between the pH 7.4 and pH 6.8 washings (*p* = 0.0234) and between the pH 6.8 and pH 7.4 washings (*p* = 0.0255). The cell numbers were labeled above the column bars. * *p* < 0.05 vs. pH 6.8.

**Figure 4 ijms-24-12783-f004:**
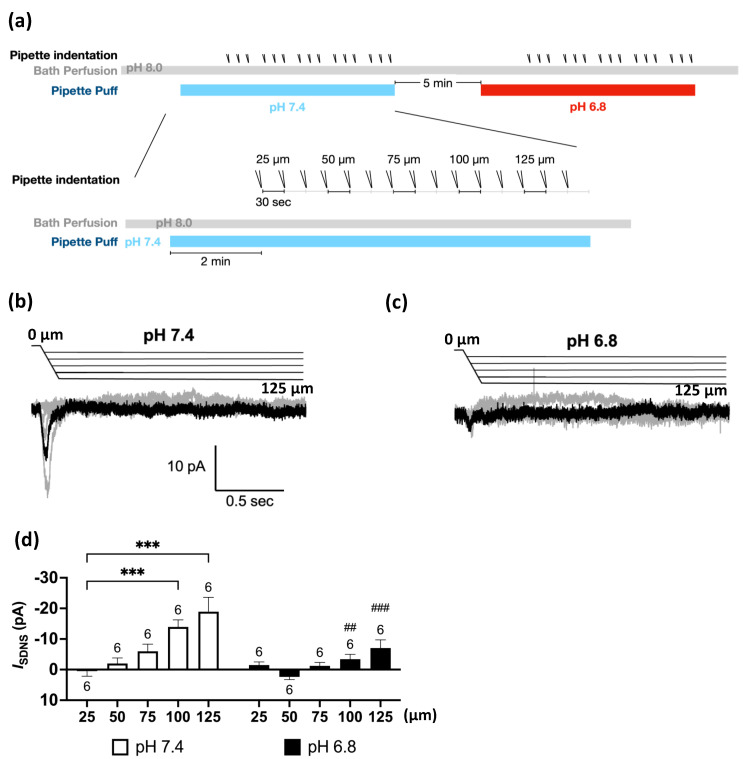
The force dependency of substrate deformation-driven neurite stretch (SDNS) currents was significantly attenuated by acidic perfusion. (**a**) Experimental protocol for testing the effect of acidosis (pH 6.8) on the force dependency of SDNS currents (*I*_SDNS_) in Pv+ DRG proprioceptors. The bath was a pH 8.0 ACSF, and the perfusion pipette contained either a pH 7.4 or pH 6.8 ACSF during the recordings of *I*_SDNS_. A series of three repeated force indentations with a depth of 25 to 125 µm was applied with the indentation pipette during the recording. The recording started at 0, and the acidic challenge began at the 5th second. The repetitive indentations stretched the neurite during the acidic challenge to generate an SDNS current; pH 7.4 (colored in light blue) and pH 6.8 (colored in red). (**b**) Force-dependent *I*_SDNS_ traces of Pv+ DRG neurons in the neutral (pH 7.4) environment. (The scale bar indicates 10 pA and 0.5 s in the trace graph.) (**c**) Force-dependent *I*_SDNS_ traces of Pv+ DRG neurons in the acidic (pH 6.8) environment. (**d**) Bar graphs of *I*_SDNS_ with different indentation depths (ranging from 25 to 125 µm) in pH 7.4 and pH 6.8 conditions. Two-way ANOVA revealed statistically significant differences in the mean test score between different depths (*F*(4, 20) = (12.53), *p* < 0.001), pH (*F*(1, 5) = (10.28), *p* = 0.02), and depth × pH (*F*(4, 20) = (7.911), *p* < 0.001). The Holm–Sidak test for multiple comparisons found that the mean value of *I*_SDNS_ was significantly different between 25 and 100 µm at pH 7.4 (*p* < 0.001); between 25 and 125 µm (*p* < 0.001); between pH 7.4 and pH 6.8 with 100 µm (*p* < 0.001); and between pH 7.4 and pH 6.8 with 125 µm (*p* < 0.001). The cell numbers were labeled above the column bars. *** *p* < 0.001, differences within the same pH group; ## *p* < 0.01 and ### *p* < 0.001, differences between the pH 7.4 and pH 6.8 groups.

**Figure 5 ijms-24-12783-f005:**
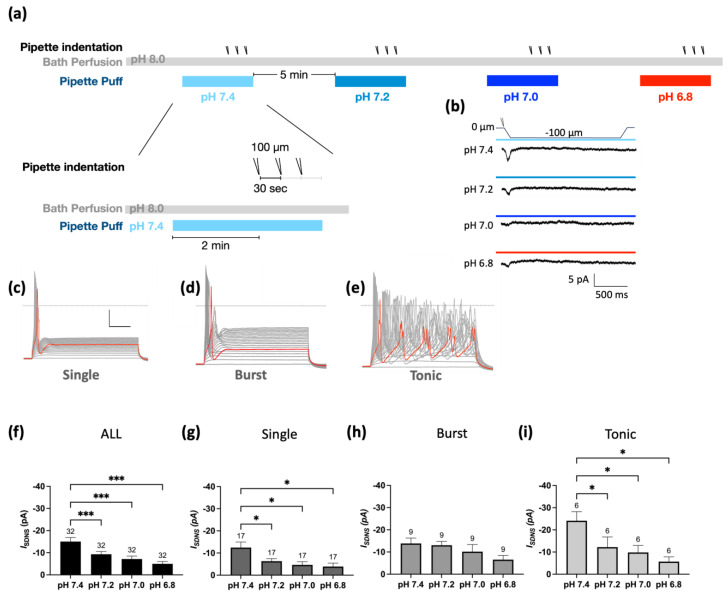
Mild acidosis pH-dependently attenuates SDNS-induced currents in Pv+ DRG proprioceptors. (**a**) Experimental protocol for testing the effect of a series of pH gradients (from 7.4 to 6.8) on SDNS-induced current (*I*_SDNS_). The bath was a pH 8.0 ACSF, and a series of pH gradients was applied using the perfusion pipette during the recording. The recording started at 0, and the acidic challenge began at the 5th second. Three repetitive indentations stretched the neurite during the acidic challenge to generate an SDNS current; pH 7.4 (colored in light blue), pH 7.2 (colored in gray-blue), pH 7.0 (colored in blue), and pH 6.8 (representing mild acidosis, colored in red). (**b**) Four normalized representative *I*_SDNS_ traces with a 100 µm depth, with three repetitive indentation traces merged for normalization. (The scale bar indicates 5 pA and 500 ms). (**c**) Representative single-firing AP traces were held at −70 mV and depolarized via 40 steps with a 10 pA increment. The gray dotted line represents zero potential level, and the red line represents a trace at the rheobase. The x- and y-axes of the scale denote 20 ms and 20 pA, respectively. (**d**) Representative burst-firing AP traces were held at −70 mV and depolarized via 40 steps with a 10 pA increment. The gray dotted line represents zero potential level, and the red line represents a trace at the rheobase. (**e**) Representative tonic-firing AP traces were held at −70 mV and depolarized via 40 steps with a 10 pA increment. The gray dotted line represents zero potential level, and the red line represents a trace at the rheobase. (**f**) pH-dependent effect of acidosis on *I*_SDNS_ of all firing types of neurons ranged from pH 7.4 to 6.8. One-way ANOVA analysis revealed statistically significant differences in the mean test score between different pH levels (*F* (2.195, 68.06) = 13.59 *p* < 0.0001). The Holm–Sidak test for multiple comparisons found that the mean value of *I*_SDNS_ was significantly different between pH 7.4 and pH 6.8 (*p* < 0.001); between pH 7.4 and pH 7.0 (*p* = 0.001); and between pH 7.4 and pH 7.2 (*p* < 0.001). (**g**) pH-dependent effect of acidosis on *I*_SDNS_ of Pv+ neurons with single-firing APs. One-way ANOVA analysis revealed statistically significant differences in the mean test score between different pH levels (*F* (1.586, 25.38) = 6.063, *p* = 0.0107). The Holm–Sidak test for multiple comparisons found that the mean values of *I*_SDNS_ were significantly different between pH 7.4 and pH 6.8 (*p* = 0.0015); between pH 7.4 and pH 7.0 (*p* = 0.0239); and between pH 7.4 and pH 7.2 (*p* = 0.0239). (**h**) pH-dependent effect of acidosis on *I*_SDNS_ of Pv+ DRG neurons with burst-firing APs. One-way ANOVA analysis revealed no statistically significant differences in the mean test score between different pH levels (*F* (1.467, 11.73) = 3.560, *p* = 0.0725). (**i**) pH-dependent effect of acidosis on *I*_SDNS_ of Pv+ DRG neurons with tonic-firing APs. One-way ANOVA analysis revealed statistically significant differences in the mean test score between different pH levels (*F* (1.974, 9.872) = 5.649, *p* = 0.02). The Holm–Sidak test for multiple comparisons found that the mean values of *I*_SDNS_ were significantly different between pH 7.4 and pH 6.8 (*p* = 0.05); between pH 7.4 and pH 7.0 (*p* = 0.05); and between pH 7.4 and pH 7.2 (*p* = 0.05). The cell numbers were labeled above the column bars in each subfigure. * *p* < 0.05 *** *p* < 0.001.

**Figure 6 ijms-24-12783-f006:**
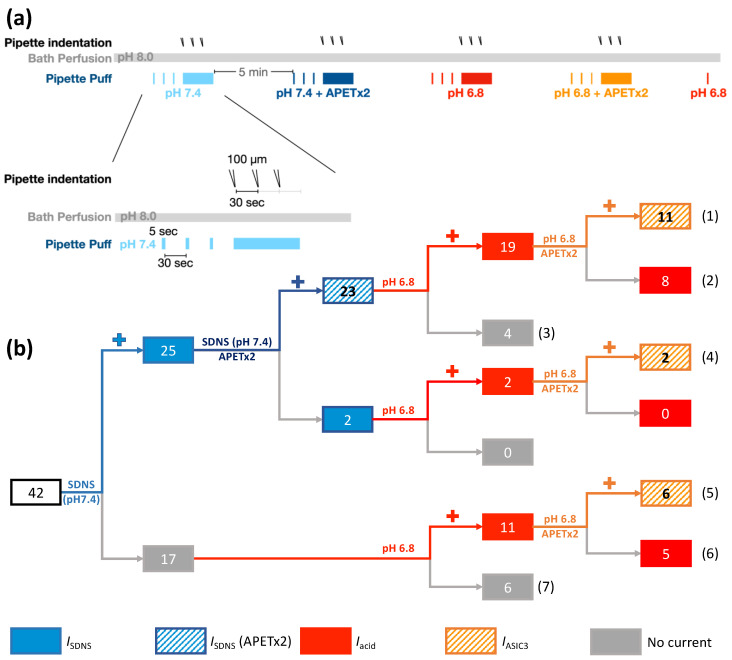
The involvement of ASIC3-containing channels in SDNS-induced currents in different Pv+ DRG proprioceptor subtypes. (**a**) Experimental design for testing the role of ASIC3 in SDNS-induced currents (*I*_SDNS_) and acid-induced currents (*I*_Acid_) during acidosis in Pv+ DRG proprioceptors. The neurons were bathed in a pH 8.0 ACSF, and different environmental conditions were established using a puff pipette with ACSF pH 7.4 (colored in light blue), ACSF pH 7.4 + APETx2 (colored in dark blue), ACSF pH 6.8 (colored in red), and ACSF pH 6.8 + APETx2 (colored in orange). In each environmental condition, the neuron first received a 5 sec puff 3 times (with a 30 sec interval) and was then subjected to three pipette indentations (with a 30 sec interval) during continuous puffing with the defined pH condition. With each pipette buffer change, the bath condition was changed back to pH 8.0 ACSF for 5 min. After all treatments, pH 6.8 ACSF was applied to confirm that APETx2 had been washed out. (**b**) Illustration of the subgrouping of Pv+ DRG neurons in response to SDNS and mild acidosis. The cell numbers were labeled in the bricks and the numbers of subgroups were labeled in the brackets.

**Figure 7 ijms-24-12783-f007:**
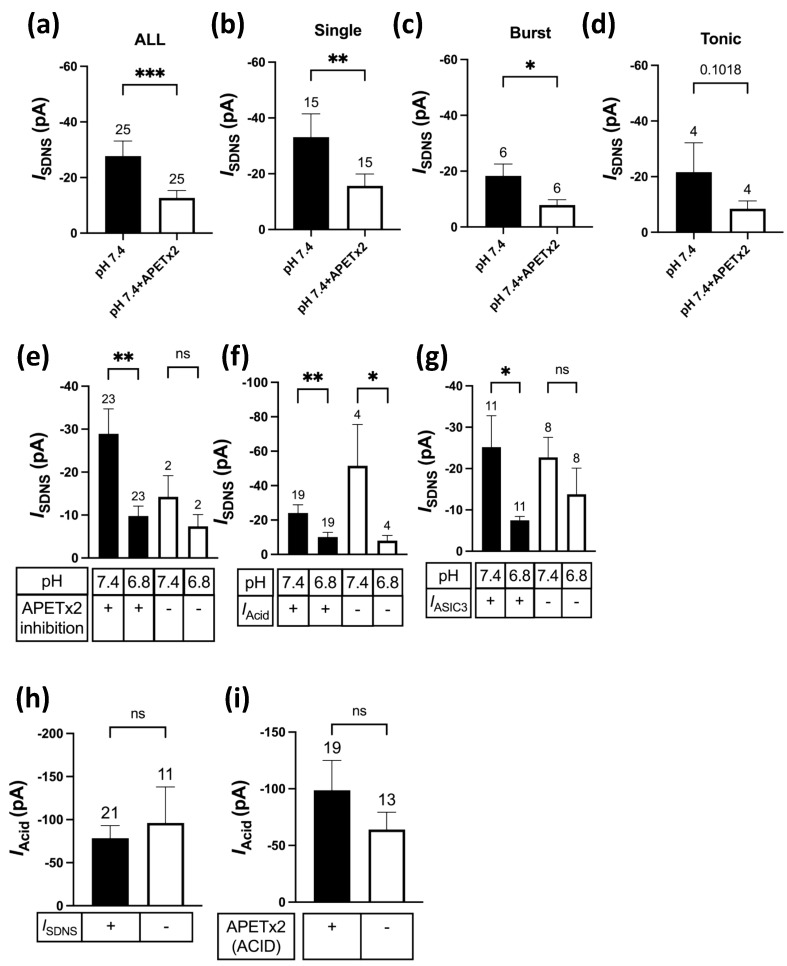
Effects of APETX2 and mild acidosis at pH 6.8 on SDNS-induced currents in different Pv+ DRG neuron subgroups. (**a**) The average peak current amplitude of the SDNS-induced currents (*I*_SDNS_) of Pv+ DRG neurons are shown. A paired *t*-test found that the mean value of *I*_SDNS_ was significantly different (*p* = 0.0002). (**b**) The average peak current amplitudes of *I*_SDNS_ (expressed in pA) of Pv+ DRG neurons with single AP are shown. A paired *t*-test found that the mean value of *I*_SDNS_ was significantly different (*p* = 0.0074). (**c**) The average peak amplitude amplitudes of *I*_SDNS_ (expressed in pA) of Pv+ DRG neurons with burst AP are shown. A paired *t*-test found that the mean value of *I*_SDNS_ was significantly different (*p* = 0.0386). (**d**) The average peak amplitude amplitudes of *I*_SDNS_ (in pA) of Pv+ DRG neurons with tonic AP are shown. A paired *t*-test found no significant difference in the mean value of *I*_SDNS_ (*p* = 0.1018). (**e**) The effect of pH 6.8 acidosis on *I*_SDNS_ between Pv+ DRG neurons with APETx2-sensitive and APETx2-resistant *I*_SDNS_. This graph indicates which SDNS current (*I*_SDNS_) was APETx2-sensitive or -insensitive. A paired two-tailed *t*-test found that the mean value of *I*_SDNS_ was significantly different between the pH levels of 6.8 and 7.4 in the APETx2 inhibition group (*p* = 0.0066). (**f**) The effect of pH 6.8 acidosis on *I*_SDNS_ between *I*_SDNS_-expressing Pv+ DRG neurons with and without acid-induced currents (*I*_Acid_). A paired two-tailed *t*-test found that the mean value of *I*_SDNS_ was significantly different between the pH levels of 6.8 and 7.4, whether in the *I*_acid_-conducted (*p* = 0.0097) or *I*_acid_-non-conducted (*p* = 0.0286) group. (**g**) The effect of pH 6.8 acidosis on *I*_SDNS_ between *I*_SDNS_-expressing Pv+ DRG neurons with APETx2-sensitive *I*_Acid_ (*I*_ASIC3_) and APETx2-resistant *I*_Acid_. A paired two-tailed *t*-test found that the mean value of *I*_SDNS_ was significantly different between pH 6.8 and pH 7.4 in the *I*_ASIC3_-containing group (*p* = 0.02). (**h**) The *I*_acid_ amplitudes of Pv+ DRG neurons with and without *I*_SDNS_ are shown. An unpaired two-tailed *t*-test found no significant difference in the mean value of *I*_SDNS_ (*p* = 0.6237). (**i**) The *I*_acid_ amplitudes of Pv+ DRG neurons with and without *I*_ASIC3_ are shown. An unpaired two-tailed *t*-test found that the mean value of *I*_SDNS_ was not significantly different, with *p* = 0.3223. The cell numbers were labeled above the column bars in each subfigure. * *p* < 0.05 ** *p* < 0.01 *** *p* < 0.001.

**Figure 8 ijms-24-12783-f008:**
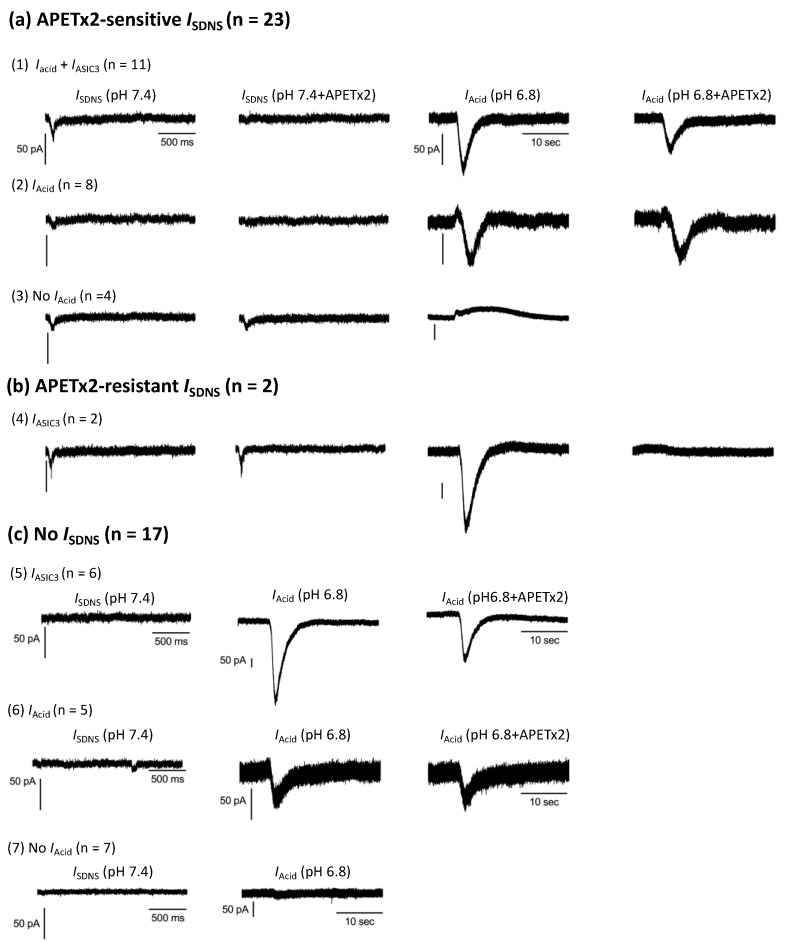
Representative traces of SDNS-induced currents (*I*_SDNS_) and acid-induced currents (*I*_Acid_) in seven groups of Pv+ DRG neurons. For the SDNS traces, the x- and y-axes of the scale bar denote 500 ms and 50 pA, respectively. For the acidic current traces, the x- and y-axes of the scale denote 10 s and 50 pA, respectively. (**a**) Pv+ DRG neurons with APETx2-sensitive *I*_SDNS_ were organized into Group (1), which expressed *I*_Acid_ + *I*_ASIC3_; Group (2), which expressed *I*_Acid_ only; and Group (3), which expressed no *I*_Acid_. (**b**) Pv+ DRG neurons with APETx2-resistant *I*_SDNS_ were organized into Group (4), which expressed *I*_ASIC3_. (**c**) Pv+ DRG neurons that did not express *I*_SDNS_ were organized into Group (5) neurons, which expressed *I*_Acid_ + *I*_ASIC3_; Group (6), which expressed *I*_Acid_; and Group (7), which expressed no *I*_Acid_.

**Figure 9 ijms-24-12783-f009:**
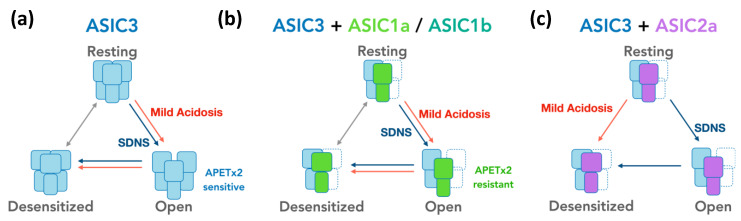
Hypothetical models illustrating the involvement of ASIC3 in the attenuation of SDNS-induced currents through mild acidosis. (**a**) A tentative model depicting the mechanistic process underlying the attenuation of SDNS-induced currents (*I*_SDNS_) in response to mild acidosis, with a particular focus on the role of acid-sensing ion channel 3 (ASIC3). The model assumes that, through mild acidosis, ASIC3-containing channels are activated by SDNS, leading to the initiation of *I*_SDNS_ in proprioceptive neurons. Mild acidosis also triggers ASCI3 homomeric channels and thus the activation of proton-gated currents (which present APETx2-sensitive acid-induced currents (*I*_Acid_)). However, due to the rapid desensitization property of ASIC3, the channel is transitioned into a desensitized state, leading to the attenuation of the *I*_SDNS_. (**b**) According to the expression of APETx2-resistant *I*_Acid_ in some Pv+ DRG proprioceptor neurons, the subunits ASIC1a and ASIC1b may contribute to neurons with *I*_SDNS_ that are less affected by mild acidosis. Pharmacological studies using APETx2, a selective ASIC3 antagonist, have shown that APETx2 effectively inhibits ASIC3 (IC_50_ = 63 nM) without affecting ASIC1a, ASIC1b, or ASIC2a, whereas ASIC1a+3 and ASIC1b+3 heteromeric channels showed low affinity to APETx2 as compared with ASIC3 or ASIC2b+3 channels. Therefore, the inclusion of the subunit ASIC1a or ASIC1b in the heteromeric channels may modify their response to the pH gradient of mild acidosis and kinetics to the attenuation of *I*_SDNS_. (**c**) In *I*_SDNS_-expressing Pv+ DRG neurons without *I*_Acid_, ASIC2a may be incorporated in the composition of the heteromeric ASIC3 channels. Previous studies have indicated that tASIC2a-containing channels have a lower half-maximum pH activation (<pH 6.0) than other combinations lacking ASIC2a [29]. This suggests that mild acidosis (with a pH level of 6.8) cannot directly activate the proton-gated current in these neurons but rather attenuates *I*_SDNS_ by transforming the resting-state channels into a desensitized state, thus reducing their responsiveness to mechanical stimuli.

**Table 1 ijms-24-12783-t001:** Action potential profile of parvalbumin-positive dorsal root ganglion neurons.

		Total			Single			Burst				Tonic			
Parameter	Unit	Mean	SEM	N	Mean	SEM	N	Mean	SEM	N		Mean	SEM	N	
Cell size	µm	34.48	1.76	100	38.19	2.44	63	29.54	2.52	23		25.93	3.08	14	*
Resting potential	mV	−53.77	0.82	105	−56.03	0.71	66	−48.30	2.41	24		−52.56	1.99	15	
Cm	pF	55.81	4.22	106	64.31	6.17	67	43.21	4.17	24		38.00	5.76	15	
Rm	MOhm	38.30	17.72	106	49.07	28.01	67	21.04	2.06	24		17.80	1.82	15	
Rheobase	pA	827.83	72.03	106	1064.18	101.30	67	441.67	56.28	24		390.00	44.77	15	
Rise slope	ms	22.68	3.21	104	25.16	4.23	67	19.34	6.56	24		16.06	6.28	13	
Decay slope	ms	−60.08	2.82	84	−61.22	3.86	51	−55.53	4.10	21		−63.19	8.55	12	
Peak amplitude	mV	80.24	1.99	105	86.51	2.41	66	70.17	3.36	24	**	68.80	4.82	15	**
Half-width	ms	5.48	2.71	103	8.01	4.34	64	1.33	0.06	24		1.33	0.11	15	

* *p* < 0.05, ** *p* < 0.01 vs. single.

## Data Availability

Not applicable.

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
