# Peer review of "Probing the Effect of Acidosis on Tether-Mode Mechanotransduction of Proprioceptors"

_ijms, 2023, doi:10.3390/ijms241612783_

Round 1
Reviewer 1 Report
In the manuscript “Probing the effect of acidosis on neurosensory mechanotransduction of proprioceptors” Cheng et al report the responses of PV-expressing DRG neurons to mechanical stimulation and to moderate acidosis. This study is based on the assumption that muscle exercise and fatigue can induce a drop in pH within the micro-environment and that this acidification may be relevant for proprioceptor signaling involving acid-sensing (e.g. ASIC) ion channels. This is an interesting issue and the present study shows that in a subgroup of DRG neurons (single and burst action potential-type neurons) whole-cell currents evoked by deformation-induced neurite stretch are inhibited by lowering pH and that this effect is susceptible to the ASIC3 channel inhibitor APETx2. Furthermore, acidosis-induced currents were detected in most PV-expressing DRG neurons and could be blocked by APETx2 in >50% of the cases. The authors use an elaborated scheme of mechanical manipulation (by substrate indentation) of cells and puff application for acidic application. Overall, the study is carefully conducted and provides interesting information on metabolic influence on mechanosensation as well as the role of ASIC3 in proprioceptive neurons.
I have only few minor comments:
Line 54: “Field” should be omitted.
Line 131: Please correct “substate deformation”.
Line 156: “59.52%“ should be given as 59% or 60%.
Line 223: “35.8%“ should be 36%.
Figure 4d: Labeling of bottom axis could include “µm”.
Author Response
Thank you for taking the time to review our article, and for your suggestions!
Based on your points, we responded as follows.

Reviewer 2 Report
Cheng and coworkers examine the effect of mild acidosis on membrane currents produced by substrate deformation-driven neurite stretch (SDNS) in cultured mouse neurite-bearing parvalbumin-expressing DRG neurons by using the whole-cell patch-clamp technique. As a result, SDNS-induced currents were found to be inhibited by mild acidosis. This inhibitory effect was greater in neurons with single and tonic APs than in neurons with burst APs in response to depolarizing pulse, and was sensitive to an ASIC3-selective antagonist APETx2. It was concluded that ASIC3 plays a pivotal role in the physiological response of proprioceptors to acidic environments and their mechanotransduction. This article has so many problems in how to present the data and how to write the manuscript. Thus, this manuscript does not appear to be written carefully. There are so many points that should be addressed and may serve to amend this manuscript, as follows:
Major points:
1. Title: this is very vague. Please describe specifically the preparation used in this study and the results obtained.
2. Fig. 1: there is no scale bar. Please amend this point.
3. Fig. 2: please put a space between value and unit in Fig. 2(b). There is no scale bar in Fig. 2(d); please amend this point.
4. Fig. 3: it will be better to write pH values in Fig. 3(b). Fig. 3(c) shows a partial reversibility, but not a reversibility. Line 428 should be revised.
5. Fig. 5: the reader does not know what “Single”, “Burst” and “Tonic” in Fig. 5(d)-(f) mean without referring to supplement Figure 1. Fig. 5 should include a figure like Supplement Figure 1(b)-(d).
6. Line 111 and others (figure legend titles and so on): the definition of SDNS has been repeatedly given. It is sufficient to give definition once. It might be good to give an abbreviation list.
7. Line 163: the concentration of APETx2 used should be given here.
8. Figures 6 and 7: raw data should be given to show how the current is suppressed by APETx2.
9. Specific values of the results obtained are repeated in figures (Figs. 3c, 4d, 5c-f, 7a-i), their figure legends and text. It is sufficient to show these values graphically. Please amend this point.
10. Throughout this manuscript, Figure legends should be placed under the figure title rather than being given separately. Please amend this point.
11. Supplement Figures 1 and 2: information is missing in their figure legends. There is no explanation about scale bars in Supplement Figure 1(b). It is unclear what the vertical and horizontal axes of Supplement Figure 2 mean. Not “um” but “μm”?  Please amend these points.
Specific points:
1. Lines 19-21: please correct English.
2. Line 43: it will be necessary to shortly explain “Runx3” and “Vglut2”.
3. Line 62: please give the definition of “ECM”.
4. Line 103: “all” is vague. Please write down the number of the nerve fibers examined specifically.
5. Line 104: it is unclear from the above sentence what “the three types” are. Please amend this point.
6. Lines 108 and 109: there are two “respectively”. Please check English.
7. Line 122: not “mOhm” but “MOhm”? Please check this point.
8. Line 130: not “neuritis” but “neurites”? Please check English.
9. Line 141: there is no “Fig. 3d” in Fig. 3. Please check this point.
10. Line 287: is “Pv+_” OK? Please check this point.
11. Line 306: not “0.02M” but “0.02 M”. Please put a space between value and unit throughout the text.
12. Line 308: please use either “hour” or “h” (see line 310) throughout the text.
13. Line 315: please write down where the companies mentioned in this manuscript, such as Abcam, are located.
14. Line 323: is “2 units of mL-1” OK? Please check this point.
15. Lines 341 and 348: is “MO” OK? Please check this point.
16. Line 350: not “monitored” but “monitor”?
17. Line 364: please use “TTX” here (see line 120).
18. Line 394: Please give evidence that the concentration of this drug is appropriate.
19. Line 505: is “Reverse” OK? Please see the above Major point (4).
20. Line 544: please check English.
21. Line 561: not “action potential” but “AP”. Since “action potential” is defined as “AP” (see line 116), in the following sentences “AP” should be used.
22. There appear to be much more scientific and writing mistakes than those pointed out above. This manuscript should be checked very carefully.
English language should be amended.
Author Response
Thank you for taking the time to review our article, and glad for your suggestions!
Based on your points, we responded as follows.

Round 2
Reviewer 2 Report
This revised manuscript has been amended in response to my comments. There are only minor points that should be considered, as follows:
1. Lines 123, 345 and 352: please use either “MOhm” (these lines) or “MΩ” (see Table 1) throughout the text.
2. Line 441: please use either “.. cerebral spinal fluid ..” (this line) or “.. cerebrospinal fluid ..” (see line 118) throughout the text.
3. There is no explanation about * and ** in Table 1. Please amend this point.
4. Fig. 5: do dotted lines in (c), (d) and (e) show zero potential level? If so, please mention this point in the legend of this figure.
5. Line 536: Fig. 7(a) shows “amplitude” but not “current”. Please amend this point.
6. Lines 540 and 542: please use either “Aps” (these lines) or “AP” (see line 538) throughout the text.
7. Line 588: please give references in “previous studies”.
8. Line 317 and others: it should be stated where the company (such as Sigma-Aldrich) from which the drug or laboratory equipment used in this study was purchased is located.
9. Line 322: not “1h” but “1 h”. Please put a space between value and unit throughout the text.
10. Line 400: is “Fig. 5” OK? Fig. 6? APETx2 does not appear to be used in Fig. 5.
The quality of English language is good.
Author Response
Thanks for your revision, again.
We have revised the manuscript based on your mentions and suggestions.
